# Enhanced Biocompatibility and Osteogenic Property of Biodegradable Zn-0.5Li Alloy through Calcium–Phosphorus Coating

Haotian Xing [1,†], Yunzhi Tang [1,†], Xinying Fa [1], Hongyun Zhang [1], Zhangzhi Shi [1,2], Shenglian Yao [1,*] and Luning Wang [1,2,*]

1 Beijing Advanced Innovation Center for Materials Genome Engineering, State Key Laboratory for Advance Metals and Materials, School of Materials Science and Engineering, University of Science and Technology Beijing, Beijing 100083, China; u202140489@xs.ustb.edu.cn (H.X.); 18663240210@163.com (Y.T.); faxinying@163.com (X.F.); m20220517@xs.ustb.edu.cn (H.Z.); ryansterne@163.com (Z.S.)
2 Institute of Materials Intelligent Technology, Liaoning Academy of Materials, Shenyang 110004, China
* Correspondence: shenglian_yao@ustb.edu.cn (S.Y.); luning.wang@ustb.edu.cn (L.W.)
† These authors contributed equally to this work.

**Abstract:** Zinc and its alloys have garnered significant attention in the field of biological implantation due to their biodegradable, osteogenic, and mechanical properties. However, the degradation of zinc and its alloys always lead to an increase in local ion concentration, and the bare metal surfaces lack biocompatibility for implantation. To address these issues, a layer of calcium–phosphorus (CaP) coating was prepared on the surface of a Zn-0.5Li alloy. The micro-structure of the coating was observed with scanning electron microscopy (SEM) and a white light interferometry microscope. The phases of the coatings were characterized through X-ray diffraction (XRD) and X-ray photoelectron spectroscopy (XPS). The bonding strength between the coating and substrate was investigated using a scratch tester with a diamond stylus, and the corrosion properties were assessed using an electrochemical method. For the evaluation of biocompatibility and osteogenic properties, MC3T3-E1 cells were cultured on the coating. Live/dead staining and proliferation tests were performed to assess cell viability and growth. Cell adhesion morphology was observed with SEM, and the level of alkaline phosphatase (ALP) in the MC3T3-E1 cells cultured on the material surface was evaluated by ALP staining and activity measurement. The CaP coating on the zinc alloy surface improved the alloy's biocompatibility and osteogenic property, and could be a promising surface modification option for a biodegradable zinc alloy.

**Keywords:** Zn-0.5Li alloy; calcium–phosphorus coating; biocompatibility; MC3T3 cells

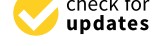



## 1. Introduction

Metallic materials are widely used in orthopedic implants due to their excellent mechanical properties. Biodegradable metal implants offer advantages over traditional non-biodegradable ones by eliminating the need for a second surgery and providing better biocompatibility. Currently, the most promising biodegradable metal implants used in orthopedics are usually magnesium-based alloys, iron-based alloys, and zinc-based alloys. Magnesium-based alloys have a fast degradation rate, with pure magnesium reported to degrade at a rate of up to 1.94 mm/y [1,2]. During the degradation process, hydrogen gas is generated and can accumulate around the tissue, leading to tissue separation from the implant, which hampers the bone healing process. Excessive hydrogen gas can also result in poor vascularization during osteogenesis and the formation of harmful gas pockets [3]. Moreover, the mechanical strength of magnesium-based alloys does not exceed 350 Mpa, and their mechanical strength decreases rapidly with in vivo degradation and cannot meet the needs of orthopedic implants [4,5]. The mechanical strength of iron-based alloys is

more than 500 MPa, and the degradation rate of iron-based materials is very low (less than 0.105 mm/y), but the corrosion products are difficult to metabolize within the body [3–5]. Therefore, neither of the above two degradable metallic materials are an optimal choice for orthopedic implants. In contrast, zinc-based alloys with good mechanical strength and degradation rates ranging from 0.17 to 0.90 mm/y could meet the needs of biodegradable metal implants [4,6,7]. Among the binary alloys of zinc-based alloys, the addition of lithium (Li) to zinc (Zn) alloys offers several notable advantages. Firstly, Li has a low density of 1.74 g/cm$^3$, which helps reduce the overall weight of the alloy. Additionally, Li is one of the most effective strengthening elements in zinc-based alloys, and only a small amount of Li addition can enhance strength [8,9]. Furthermore, both of the zinc and lithium ions released from the Zn-0.5Li alloy have beneficial effects on bone growth [8]. Multiple references reported that zinc ions play a crucial role in stimulating bone formation and mineralization, and inhibiting osteoclast formation [10]. It was also reported that Li$^+$ can enhance bone formation and improve bone mass in mice [11,12]. However, the burst increase in Zn$^{2+}$ may occur during the Zn-0.5Li alloy's degradation in body fluids, resulting in a localized high Zn$^{2+}$ concentration. A high concentration of Zn$^{2+}$ always causes cytotoxicity and excessive inflammation [13–15]. For example, Tong et al. found that excessive zinc ions significantly affected the proliferation of MC3T3-E1 cells and increased the ratio of erythrocyte rupture to apoptosis [15]. Jablonská et al. observed that extracts of an untreated Zn-1.5 Mg alloy specimen caused 100% cytotoxicity (determined as zero metabolic activity) in L929 cells, and the metabolic activity of the cells strongly correlated with Zn$^{2+}$ concentration in the extracts [16]. Therefore, the release of Zn$^{2+}$ from the alloy should be controlled to an acceptable concentration [17].

Surface coatings are an effective method to control ion release and to improve the biocompatibility of metals [18,19]. Generally, compared to other coating materials [1], calcium–phosphorus coatings with great osteoconductivity and osteoinductivity are undoubtedly ideal bioinorganic coating materials for bone implantation [20–23]. In terms of elemental composition, the CaP coating is the closest to the elements of bone. Ca$^{2+}$ and PO$_4{}^{3-}$ both play important roles in the process of osteogenesis. Additionally, the complex structure of its surface is more conducive to cell adhesion, growth, diffusion, and cell differentiation. Some studies have shown that the microstructure of the CaP coating also has a certain role in the differentiation of osteoblasts [23]. It was reported that a Zn-1.5 Mg alloy was treated in a supersaturated calcium–phosphorus solution at 37 °C for 14 days, which resulted in the formation of a protective layer of calcium-rich phosphate on the alloy surface, and that cell viability and adhesion were significantly enhanced when U-2 OS cells were cultured on the calcium phosphate-protected film [16]. In another study, CaP coatings were prepared by high-frequency magnetron sputtering, and it was found that bone marrow mesenchymal stem cells (MSCs) adhered well to the coating surface and showed good proliferation [24]. In an in vivo study of CaP coatings prepared on zinc alloy surfaces, no significant inflammatory or adverse reactions were observed around the implants. New bone formation could be observed surrounding the implant at week 4 and plenty of osteocytes were seen in the new bone tissue, and the new bone mass in CaP-coating groups was significantly higher than that of pure Zn [13].

For the preparation of CaP coatings on alloy surfaces, various physical deposition methods have been used, such as thermal spraying, matrix-assisted pulsed laser evaporation, magnetron sputtering, etc. Generally, these physical deposition methods often require harsh process conditions [25]. For example, atmospheric plasma spraying involves injecting calcium phosphate powder into a plasma jet, which operates at extremely high temperatures in the range of ten thousand degrees. Such high temperatures can cause incongruent melting of calcium phosphate particles, leading to structural modifications, uncontrolled phase changes, and chemical decomposition [26–28]. Wet chemical methods, such as sol–gel, micro-arc oxidation, electrochemical deposition, etc., have been engaged to prepare CaP coatings due to the advantages of short coating formation times, applicability to irregular metal surfaces, low cost, and reduced environmental impact [25,29].

Therefore, the chemical conversion method is widely used to prepare CaP coatings on the surfaces of biomedical metals. During the in situ growth process of coating, chemical bonds could form between a metal substrate and coating layer, enhancing adhesion of the coating to the substrate [20]. It was reported that a Ca-P-coated Zn-Mg alloy was obtained by immersion in a $CaCl_2$ solution, and $Zn^{2+}$ release could be modulated. MC3T3-E1 cells directly seeded on a Ca-P coating exhibited excellent attachment and spreading. Compared with the bare alloy, the Ca-P coating improved corrosion resistance, antibacterial activity, and biocompatibility [20].

In this study, CaP coatings were prepared on the surface of zinc alloys by the chemical conversion method. The surface morphology, microstructure, and chemical composition of the coatings were characterized, and the bond strength between the coatings and the substrate was investigated by micron scratch testing. The effect of the CaP coating on the degradation rate of the Zn-0.5Li alloy was also explored by simulated body fluid (SBF) immersion experiments. MC3T3-E1 cells were cultured on the surface of the samples to explore the biocompatibility of the Zn-0.5Li alloy, as well as to characterize osteogenic activity by ALP staining and activity measurement.

## 2. Materials and Methods

### 2.1. Material and Surface Pretreatment

The matrix material used in this study was rolled Zn-0.5 Li alloy (processed by Jinzhou Fushibo New Material Research Co., Jinzhou, Liaoning, China). The samples were cut into discs with a diameter of 10 mm and a thickness of 1 mm. These discs underwent a sanding process using 80#, 400#, and 1000# sandpaper to remove the surface quenching layer and surface defects (as shown in Scheme S1), and then were ultrasonically rinsed with acetone, ethanol, and deionized water for 15 min. Following the cleaning process, the samples were dried for further analysis.

### 2.2. Coating Preparation

The Zn-0.5Li alloy discs were submerged in a phosphating solution (composition: 23.6 g/L Ca $(NO_3)_2$-$4H_2O$, 34.2 mL/L $H_3PO_4$ (85%)), and then the pH of the phosphating bath was adjusted to 2.8 with NaOH at 37 °C. The samples treated with the solution for 5 min, 10 min, 15 min, and 60 min were named CaP-5, CaP-10, CaP-15, and CaP-60, respectively. After coating preparation, all coated samples were rinsed with deionized water and dried for further characterization.

### 2.3. Coating Characterization

The surface morphology and microstructure of the coatings were observed via a field emission scanning electron microscope (SEM, Regulus-8100, Hitachi, Tokyo, Japan) equipped with an energy dispersive spectrometer (EDS, Oxford Instrument Inca X-Maxn-Sight). The three-dimensional structure was observed by a white light interferometer microscope (WLIM, Contour GTK, Bruker, Billerica, MA, USA). The phase compositions of the coatings were detected by an X-ray diffractometer (XRD, SmartLab, Rigaku, Tokyo, Japan), at a scanning speed of 4° per min between 5° and 80°, and Cu-K$\alpha$ radiation. The bonding strength between the coatings and substrates was tested using a scratch tester (N2100, Nanovea, Irvine, CA, USA) with a diamond stylus. The scratches were conducted under a continuously increasing load from 0 to 100 N at a loading rate of 100 N min$^{-1}$ and the scratch length was set as 5 mm. The load at which the coating was totally peeled off from the Zn-0.5Li substrate was defined as the critical load (Lc) and the moment was recorded by an acoustic emission signal. Each specimen was scratched three times, and a quiet environment was necessary during the scratch operating in order to avoid noise pollution.

### 2.4. Electrochemical Measurements

Electrochemical measurements were performed by an electrochemical analyzer (Mod-uLab XM, Ametek, San Diego, CA, USA). A three-electrode-cell set-up was used, in which

the specimen, a saturated calomel electrode (SCE), and a platinum sheet were used as the working electrode, reference electrode, and counter electrode, respectively. Open circuit potential (OCP) measurements were recorded after a 30 min immersion. Potentiodynamic polarization (PDP) tests were conducted in a range of $\pm 0.3$ V vs. OCP at a constant scan rate of 1 mV s$^{-1}$. Electrochemical impedance spectroscopies (EISs) were carried out at an OCP of 5 mV sinusoidal amplitude in the frequency range of $10^5$ Hz to $10^{-2}$ Hz. The impedance data were analyzed with ZSimpWin(version 3.30) software and fitted to the equivalent curves. Linear polarization resistances (LPRs) of the specimens were measured at $\pm 20$ mV and OCP with a potential sweep rate of 0.167 mV s$^{-1}$. All experiments were conducted in SBF. The order, amounts, weighting containers, purities, and formula weights of reagents for preparing 1000 mL of SBF can be found in Ref [30,31].

### 2.5. Immersion Test in SBF

The early corrosion behavior of specimens was investigated by immersion in SBF at 37 °C with an immersion ratio of 0.2 cm$^2$/mL. The soaking time was 1 d, 3 d, 5 d, 7 d, 14 d, 21 d, and 28 d, respectively. The SBF solution was changed every other day during the immersion experiment. The release amounts of $Zn^{2+}$ and $Li^+$ in SBF were evaluated with inductively coupled plasma atomic emission spectrometry (ICP-AES, OPTIMA 7000DV, PerkinElmer, Waltham, MA, US). Three parallel specimens were tested for each group. The corrosion morphology of immersed samples was observed by SEM. XRD with Cu-K$\alpha$ radiation at a scanning speed of 4°/min was employed to analyze corrosion products.

### 2.6. In Vitro Cell Culture
#### 2.6.1. Cell Proliferation

MC3T3-E1 cell proliferation on the surface of samples was detected by the CyQUANT NF Cell Proliferation Assay Kit (TREK Diagnostic Systems, OH, USA). After the cells were cultured on different surfaces (tissue culture plate (TCP), untreated, CaP-5, CaP-10, CaP-15, and CaP-60) for 1 d, 3 d, and 5 d, the amount of deoxyribonucleic acid (DNA) was detected to quantify the cell number. Generally, according to the instructions of the CyQUANT kit, 11 mL of 1× HBSS buffer was prepared by diluting 2.2 mL of 5× HBSS buffer with 8.8 mL of deionized water, and then a 1× dye-binding working solution was prepared by adding 22 μL of CyQUANT NF dye reagent to 11 mL of 1× HBSS buffer. After cell lysis, 200 μL of 1× dye-binding working solution was added to each well and incubated for 30 min at 37 °C in the dark. Finally, the fluorescence intensity of the solution was measured by a multifunctional enzyme marker with an excitation wavelength of 485 nm and an emission detection wavelength of 530 nm.

#### 2.6.2. Live/Dead Cell Staining

MC3T3-E1 cells ($5 \times 10^4$ cells/mL) were seeded on different specimens in 48-well plates. After 2 days of culture, the cells were washed with PBS and then stained with propidium iodide (PI) and Calcein-AM reagents (CA1630, Solarbio, Beijing, China) at 37 °C for 15 min. The working solution concentrations of Calcein-AM and PI were 2 μM and 4.5 μM, respectively. After washing the stained cells with PBS, the cells were observed by fluorescence microscopy (Axio Vert.A1, Zeiss, Oberkochen, Germany).

#### 2.6.3. Cell Morphology

After 48 h of cell culture, the samples were rinsed with PBS three times and fixed with 2.5% glutaraldehyde at 4 °C for 2 h. Then, the specimens were dehydrated with graded ethanol (30%, 50%, 60%, 70%, 80%, 90%, 95%, 100%) for 15 min in turn. Finally, the specimens were soaked in acetone for 30 min and dried with a carbon dioxide critical dryer. Samples were imaged by SEM (SEM, Regulus-8100, Hitachi, Tokyo, Japan) after being sputter-coated with a layer of Au.

#### 2.6.4. Alkaline Phosphatase Measurement

After MC3T3-E1 cells were cultured on the sample surface for 7 days, the samples were stained using a BCIP/NBT alkaline phosphatase color development kit (Beyotime, Shanghai, China) according to the manufacturer's protocol. Then the samples were observed by a microscope. For ALP activity measurement, cells were cultured on different surfaces for 7 days, then washed with PBS, and finally lysed in RIPA cell lysis buffer (Beyotime, China). ALP activity in the lysate was measured using an ALP testing kit (Jiancheng Bio-engineering Research Institute of Nanjing, China). ALP activity was normalized by the total protein content determined with a BCA Assay Kit (Beyotime, China), according to the company's guidelines. The experiments were carried out in quadruplicate.

### 2.7. Statistical Analysis

All experiments were carried out in triplicate and the data are presented as the mean $\pm$ standard deviation. Statistical comparison was performed using ANVOA, and $p$-values $< 0.05$ are statistically significant.

## 3. Results

### 3.1. Morphology, Structure, and Properties of CaP Coatings

Figure 1a,b,e,f,i,j,m,n show the microstructure of the coating layer on the Zn-0.5Li alloy treated with a phosphating solution for different times. It could be observed that a layer of micro-irregular particles was deposited on the surface of the Zn-0.5Li alloy. The high-magnification images revealed that particles, with an average diameter of approximately 4 μm, formed connections with each other, effectively covering the surface. However, it was observed that only the CaP-5 group did not completely coat the metal surface. With the reaction time increased, micro-particles composed on the coating layer became closer to cover the surface, and cracks were observed on the coating layer when the reaction time reached 60 min. Further observation of the coating layer bonding with the Zn-0.5Li alloy substrate is shown in the cross-section images in Figure 1c,g,k,o, and element (P, Ca, O, and Zn) distribution along the black arrow is shown in Figure 1d,h,i,p, respectively. In the CaP-5 group, micro-particles only covered part of the alloy surface, and the average thickness was about 5 μm. In the CaP-10, CaP-15, and CaP-60 groups, the average thickness of the coating layer was 8 μm, 11 μm, and 15 μm, respectively. There were no cracks between the coating layer and the substrate in the CaP-5 and CaP-10 groups. However, in the CaP-15 and CaP-60 groups, significant cracks between the coating and the alloy substrate were observed, and the size of the cracks increased with the thickness of coating. Although the amount of zinc atoms slumped from the alloy substrate to the coating layer, the zinc element was also obviously detected in the coating layer. The alloy substrate did not contain calcium (Ca), phosphorus (P), or oxygen (O), while all three elements were significantly increased in the coating layer.

The surface roughness of the coating layer was evaluated by a white light interferometry microscope, and the 3D images are shown in Figure 2a–e with the quantified results shown in Figure 2f. Compared to the untreated zinc alloy surface, the micro-particles deposited on the surface could increase the roughness. The depositing of micro-particles increased the thickness of the layer, and the surface roughness decreased in the CaP-15 and CaP-60 groups. The highest roughness value of 0.575 μm was observed for the CaP-10 sample, while the lowest roughness value of 0.158 μm was observed for the CaP-60 sample.

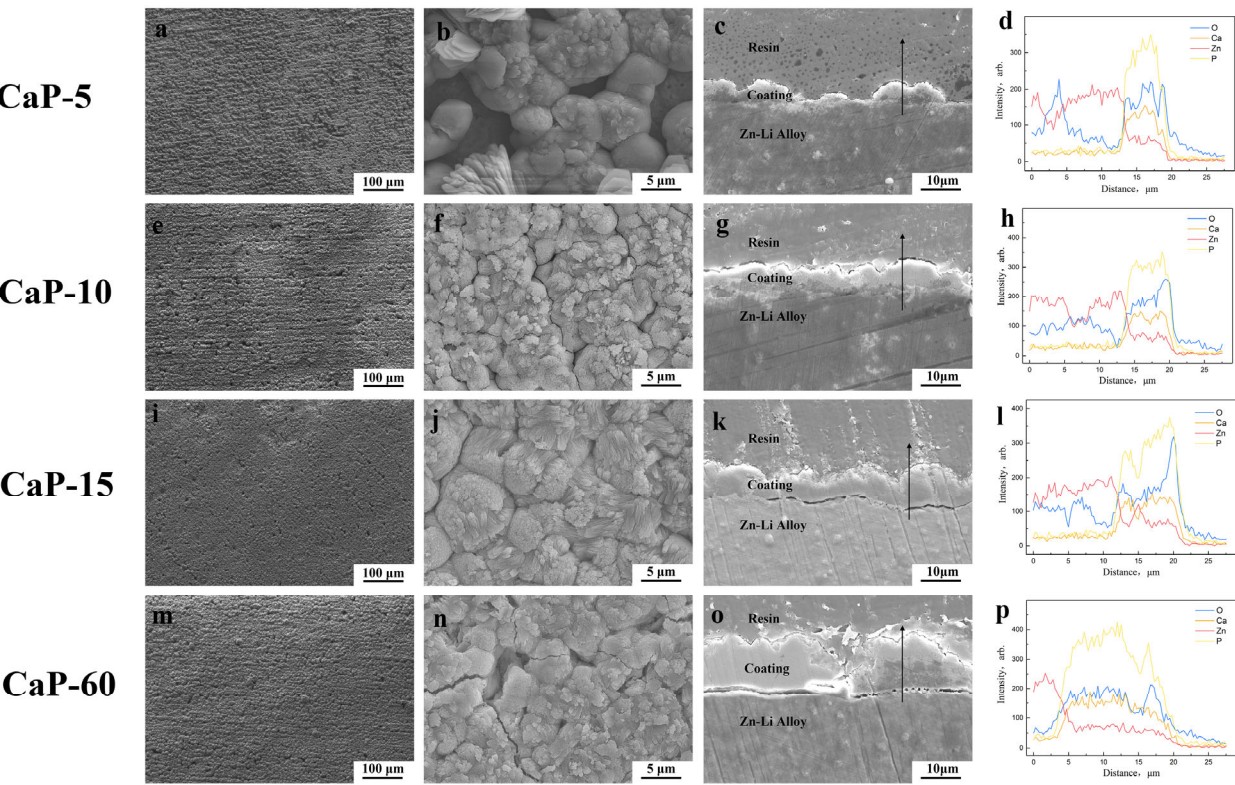

**Figure 1.** SEM micrographs of CaP coatings at different magnifications and element distribution obtained with EDS: (**a**–**d**) CaP-5; (**e**–**h**) CaP-10; (**i**–**l**) CaP-15; (**m**–**p**) CaP-60.

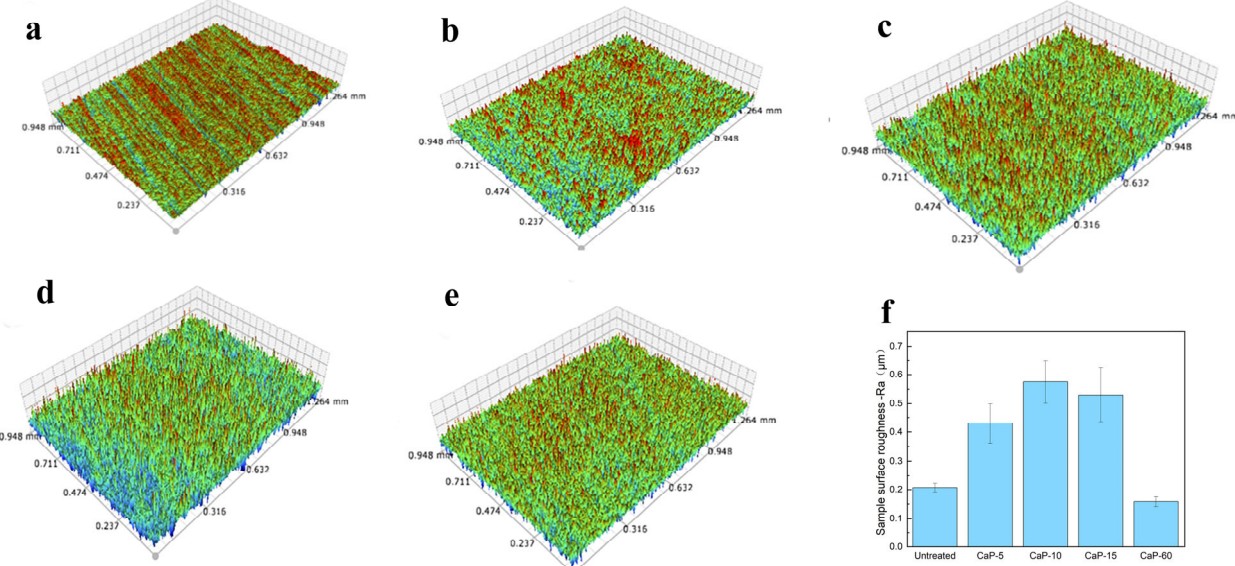

**Figure 2.** Three-dimensional morphology and surface roughness profiles of CaP coatings: (**a**) untreated; (**b**) CaP-5; (**c**) CaP-10; (**d**) CaP-15; (**e**) CaP-60; (**f**) sample surface roughness (μm).

The XRD results of bare Zn-0.5Li alloy and coated samples are shown in Figure 3A(a). For the bare Zn-0.5Li alloy, pure Zn phase and β-LiZn$_4$ with diffraction peaks of 2θ angles at 21.91°, 41.4°, and 42.6° were detected. For further analyses of the phases in the coating layer, a high-magnification image of the diffraction peaks from 5 ° to 45° is shown in Figure 3A(b). In addition to the diffraction peaks detected for matrix Zn and β-LiZn$_4$, peaks corresponding to β-Hopeite (Zn$_3$(PO$_4$)$_2$-4H$_2$O, JCPDS# 37-0456), Scholzite (CaZn$_2$(PO$_4$)$_2$-

$2H_2O$, JCPDS# 35-0495), and $ZnLiPO_4$ (JCPDS# 83-0263) were observed. To further identify components of the coating, XPS results are shown in Figure 3B. The XPS results showed the presence of P, O, Ca, and Zn elements in the coating layer. Figure 3B(a) shows the spectra of P2p. The P $2p_{1/2}$ peak and P $2p_{3/2}$ peak positions were located at 140.3 eV and 133.4 eV, respectively. This indicated that the P element in the coating existed in the form of $PO_4^{3-}$, $H_2PO_4^{-}$, and $HPO_4^{2-}$. The spectra of O1s in Figure 3B(b) showed a peak at 531.5 eV, which corresponded to $PO_4^{3-}$. The spectra of Ca 2p in Figure 3B(c) showed the binding energies of $Ca2p_{3/2}$ and $Ca2p_{1/2}$ were 347.65 eV and 351.25 eV, respectively, which corresponded to $Ca_3(PO_4)_2$. Figure 3B(d) shows the spectra of Zn 2p. The Zn $2p_{1/2}$ peak position was located at 1022.3 eV and 1045.3 eV, which fits with the Zn ion in ZnO and $Zn_3(PO_4)_2$-$4H_2O$, respectively.

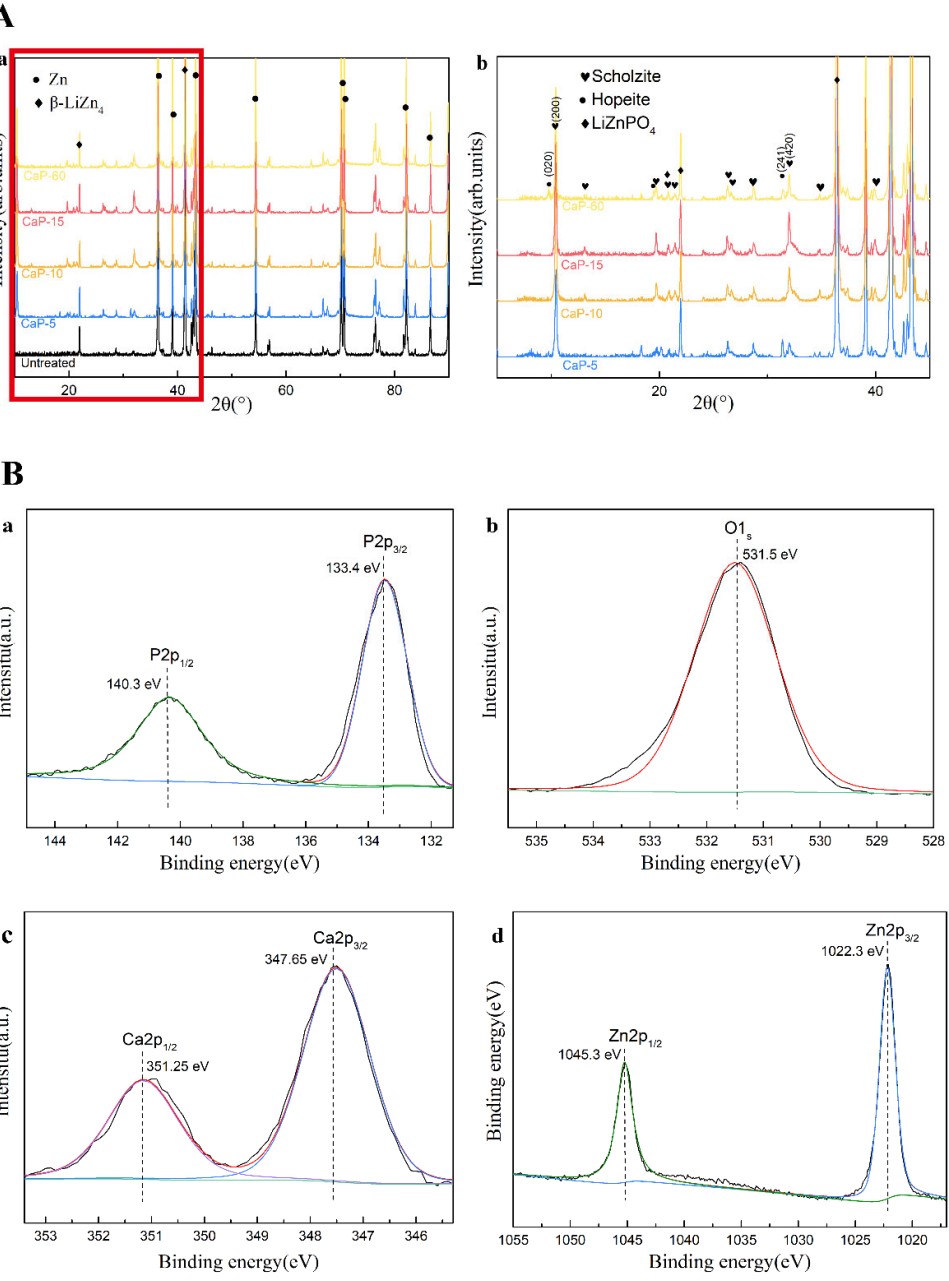

**Figure 3.** Analysis of XRD and XPS results of CaP coatings: (**A**) XRD patterns of the CaP coating, where (**A**(b)) is the magnification of the red area in (**A**(a)); (**B**) XPS spectra of CaP-60: (**B**(a)) P 2p; (**B**(b)) O 1s; (**B**(c)) Ca 2p; and (**B**(d)) Zn 2p.

One of the most critical parameters for coatings is the strength of adhesion to the substrate. In this work, the bonding strength was obtained from a micron scratch test, which is performed by applying compressive stress to the front of the diamond stylus, and then the critical load ($L_C$) was obtained. In Figure 4, representative acoustic emission (AE) signals of coating layers under the test load are shown. The average $L_C$ values from the acoustic emission signals of the CaP-5, CaP-10, CaP-15, and CaP-60 samples were measured at 20.50 N, 10.60 N, 9.12 N, and 7.74 N, respectively. It can be concluded from the image that the failure occurred according to the adhesive mechanism. No cracks are observed near the scratch for all the coating groups. The scratch borders were well defined and abrupt. The coating deterioration exhibited a ductile character with a low percentage of generated debris.

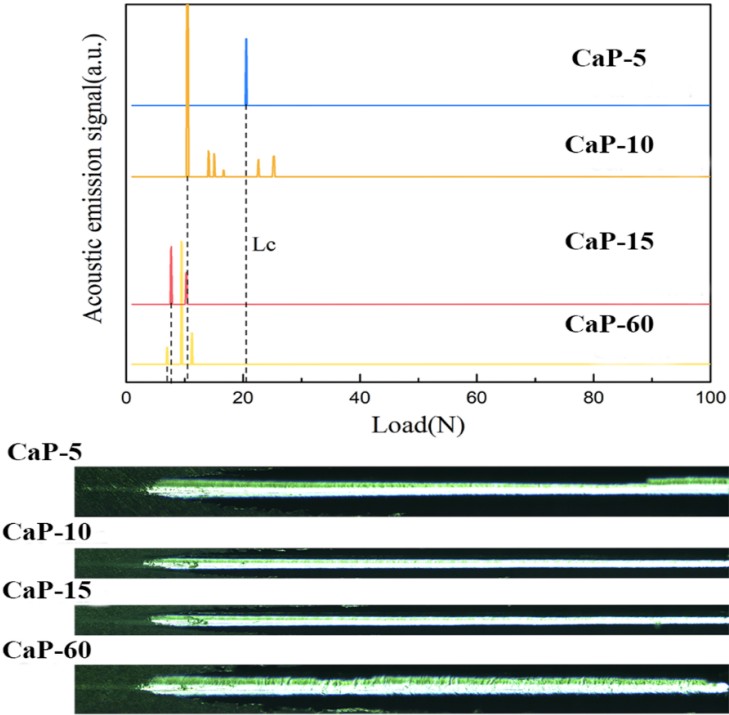

**Figure 4.** Acoustic-emission load curves and optical images of scratches on CaP coating surfaces.

*3.2. Electrochemical Test*

Figure 5a,b show the EIS spectra of the samples coated with different thicknesses of CaP. As shown in Figure 5a, Nyquist plots of the samples showed a quasi-semicircular arc, and the diameter of the semicircles corresponds to the charge transfer resistance ($R_{CT}$). With increasing thickness of the coating layer, the radius of its impedance arc also increased, and that indicated better corrosion resistance. It was observed that the CaP-60 group showed the highest corrosion resistance among all the groups. All the samples showed a rising impedance modulus ($|Z|$) at a low frequency, while the impedance decreased at a high-frequency range (Figure 5b). Figure 5c demonstrates that as the thickness of the sample coating increased, the corresponding phase angle became wider in the high-angle-phase-angle range. Additionally, the phase angle of the coated samples was larger in the mid-frequency range. These observations indicated that the coating was characterized by a large resistance and a small capacitance, which contributed to enhance corrosion resistance. The equivalent circuit ($E_C$) for the coated samples in the electrochemical experiments shown in Figure 5d was established by referring to You et al. [17]. The model fully considered the various parameters, such as solution resistance ($R_s$), electrical resistance, coating capacitance ($R_1$ and $CPE_1$), and charge transfer process ($R_{ct}$ and $CPE_2$) in the experiment. It also considered the resistance of the interfacial bilayer ($Rd_l$), the resistance generated by the CaP layer and the corrosion layer ($R_1$), and the diffusion resistance ($G$).

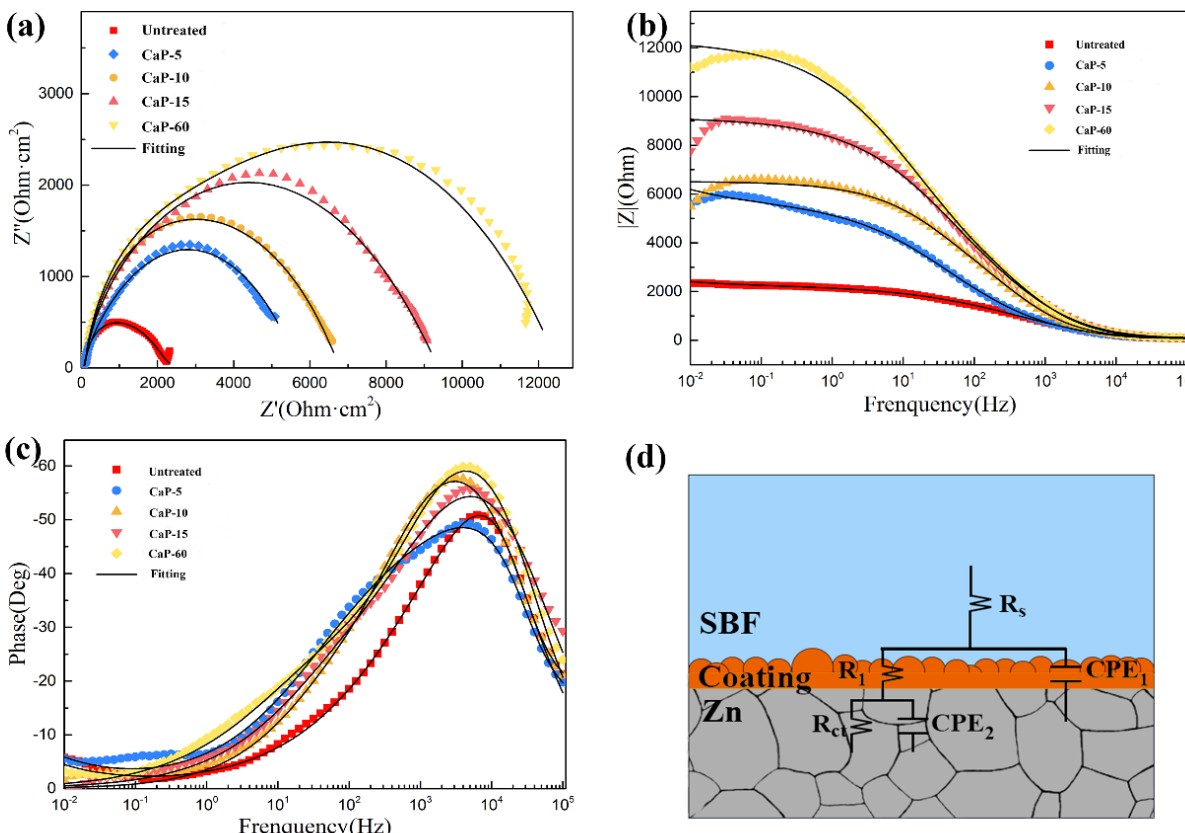

**Figure 5.** EIS results of Zn-0.5Li alloy and CaP-coating samples in SBF solution: (**a**) Nyquist plot, (**b**) Bode plot: |Z| vs. frequency; (**c**) Bode plot: phase angle vs. frequency; (**d**) illustration of the equivalent circuit of CaP coating in electrochemical measurements.

*3.3. In Vitro Degradation Behavior*

The SEM images in Figure 6A(a–e) show the corrosion morphology of untreated Zn-0.5Li alloy and CaP-coated samples which were immersed in SBF solution for 28 days in vitro. There were obviously irregular nanoparticles depositing on the surface of all the samples, and the distribution of the corrosion products was random on the surface as the high-magnification images (insert of each SEM image) show. The morphology of the corrosion products is similar for all the samples. Notably, irregular pitting pits appeared everywhere on the untreated Zn-0.5Li alloy (Figure 6A(a)), while there were no pitting pits on all the CaP-coating surfaces. The coating layer on the alloy surface could preserve the metal substrate from non-uniform corrosion well and inhibit the appearance of irregular pitting pits. To further identify the corrosion products, XRD spectra of the samples are shown in Figure 6A(f). As the XRD results showed, in addition to the coating layer phase, ZnO (JCPDF #36-1451), $Zn(OH)_2$ (JCPDF #20-1435), and $Zn_5(CO_3)_2(OH)_6$ (JCPDF #19-1458) diffraction peaks were detected on the surfaces of the coated samples. All the products were typical corrosion products of the Zn-0.5Li alloy. Thus, the CaP-coating layer didn't hinder the degradation of the Zn-0.5Li alloy, and preserved the metal substrate from non-uniform corrosion well.

The release behavior of ions is shown in Figure 6B. As shown in Figure 6B(a), the release concentration of zinc ions in SBF solution was high for the first 5 days, and then decreased to a stable low concentration in the CaP-10, CaP-15, and CaP-60 groups. The CaP-5 group showed similar zinc ion release behavior with the untreated Zn-0.5Li alloy, and the zinc ion release rate was high for the first 5 days and then increased at 20 days with a decrease between 5 days and 15 days. Even in the first 3 days, the releasing $Zn^{2+}$ concentration of CaP-5 was higher than the untreated surface. The burst increase of

$Zn^{2+}$ concentration in the untreated surface and the CaP-5 group should be caused by non-uniform corrosion with the formation of pitting-corrosion pits on the alloy surface. Different from $Zn^{2+}$ release behavior, the CaP-coated samples showed similar $Li^+$ release concentrations with the untreated Zn-0.5Li alloy (Figure 6B(b)).

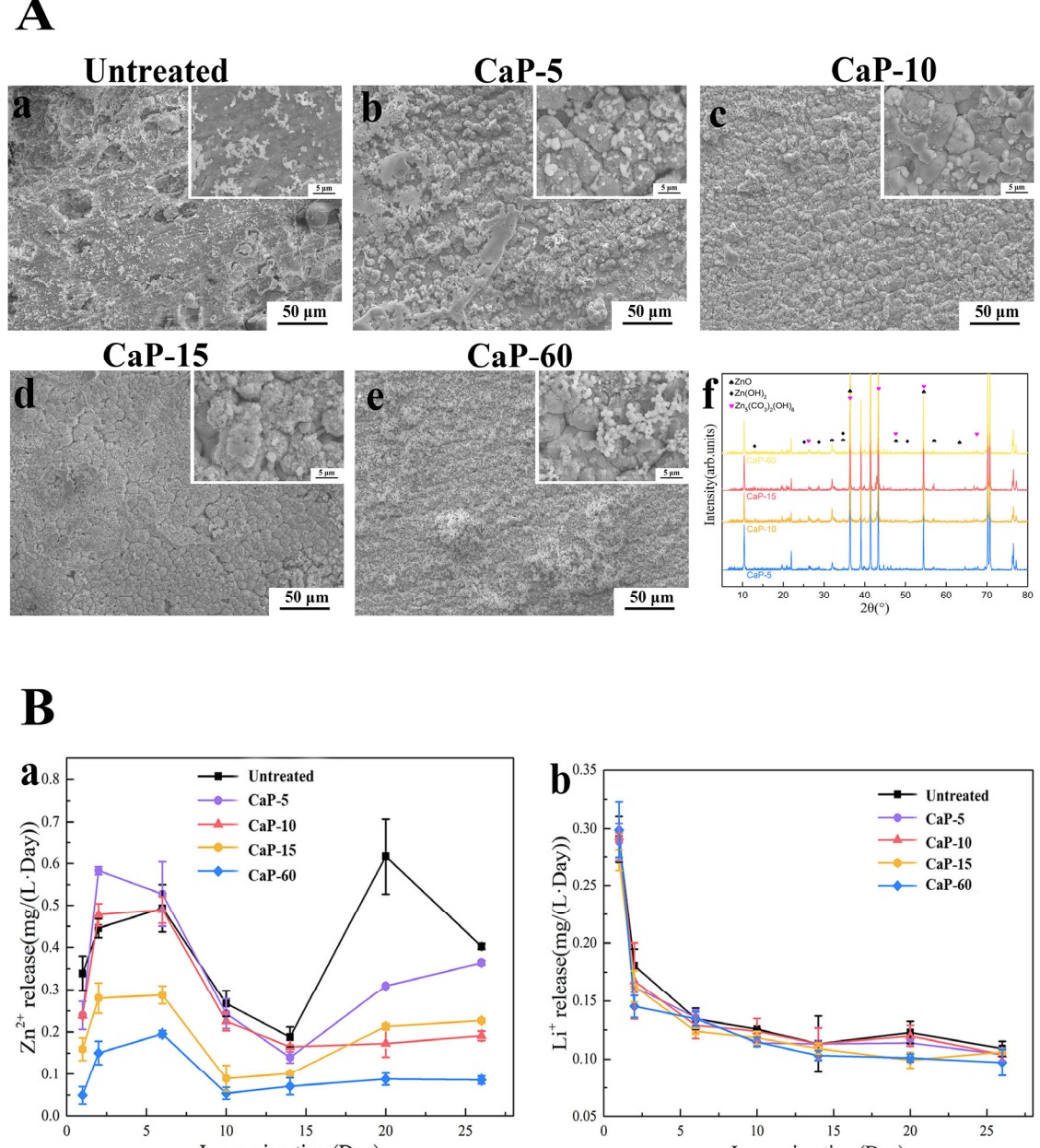

**Figure 6.** CaP-coating in vitro degradation behavior: (**A**) morphology of the corrosion surface observed with SEM and the corrosion products characterized with XRD pattern (**A**(f)); (**B**) ion release behavior with/without the CaP coating.

### 3.4. Cell Viability, Adhesion, and Proliferation

The viability of MC3T3-E1 cells cultured on bare Zn-0.5Li alloy and CaP-coating samples for 2 days was characterized by live/dead staining. As shown in Figure 7A(a–f), almost no live cell adhered on the surface of bare Zn-0.5Li alloy, while the number of adhesive live cells increased with the addition of the CaP-coating layer on the alloy surface. Almost all the cells cultured on CaP-15 and CaP-60 surfaces were alive and showed a spreading morphology. The live cell density of CaP-15 and CaP-60 groups was high and

like the TCP group. The cells cultured on the CaP-5 surface showed a low cell density and spherical morphology. For the CaP-10 group, most of the cells were alive and showed a spreading morphology, but some of the cells showed spherical morphology as the arrows show. Thus, the CaP coating on the Zn-0.5Li alloy surface could improve cell viability and spreading adhesion. Figure 7B shows the cell proliferation of MC3T3-E1 cells cultured on different surfaces for 1 d, 3 d, and 5 d with a cell proliferation assay kit (CyQUANT NF Cell Proliferation Assay Kit). On the bare Zn-0.5Li alloy surface, there was no significant cell proliferation, and the cell number remained at a low level from 1 d to 5 d. While for the CaP-coating surfaces (CaP-5, CaP-10, CaP-15, and CaP-60), the cells showed significant proliferation, especially for the CaP-10, CaP-15, and CaP-60 groups. To further observe cell adhesion on the CaP coating, SEM images are shown in Figure 7C after 2 days of cell culture. The cells cultured on the untreated Zn-0.5Li alloy surface showed no cell spreading and remained a spherical shape (Figure 7C(a,f). Although the cell spreading area was different on the CaP-coating surfaces (Figure 7C(b–e)), the cells showed good contact with CaP particles as shown in the high-magnification images (Figure 7C(g–j)).

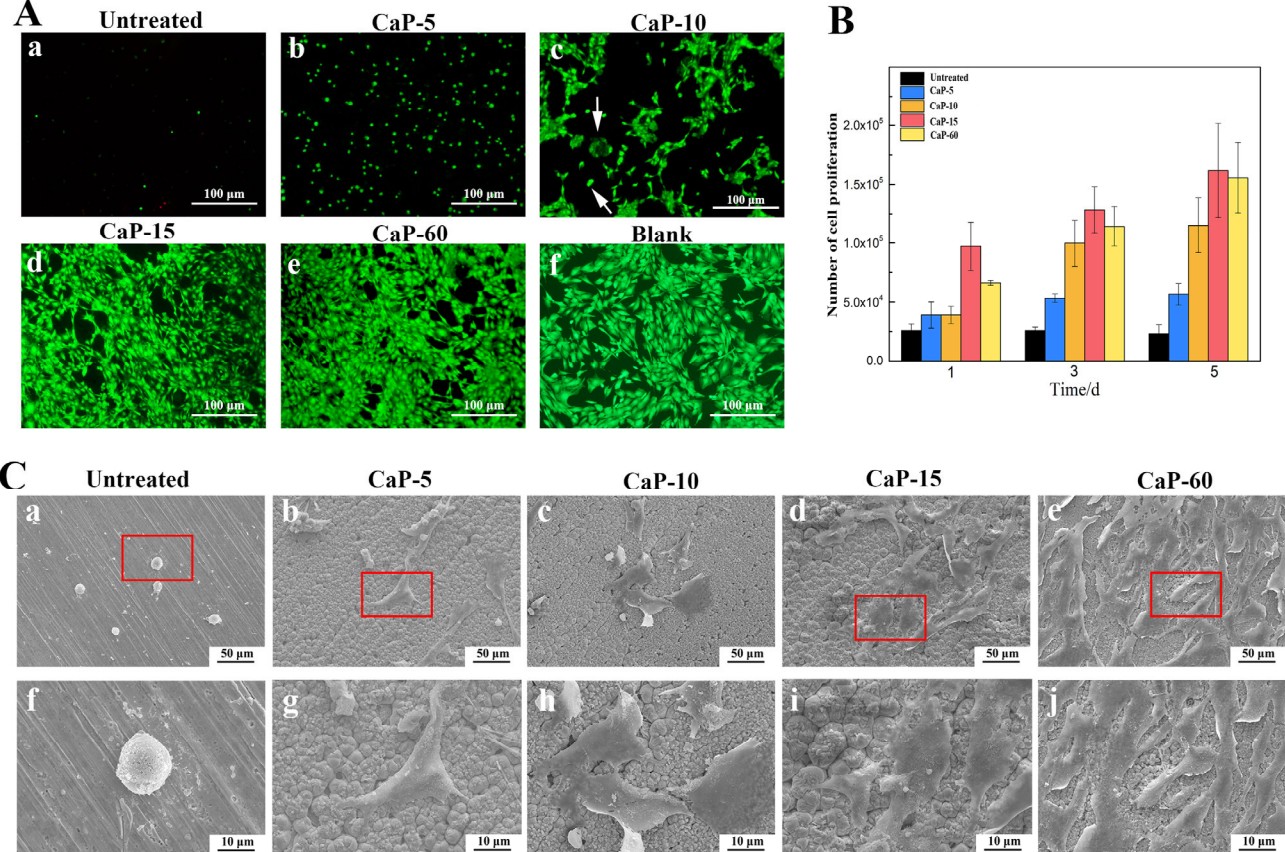

**Figure 7.** MC3T3-E1 cell viability, proliferation, and adhesion morphology. (**A**) Live/dead staining of cells cultured on different surfaces for 2 days; (**B**) cell proliferation; (**C**) SEM images of cell morphology on different surfaces.

### 3.5. Osteogenic Activity

ALP staining was processed for MC3T3-E1 cells cultured on untreated Zn-0.5Li alloy and CaP-coating surfaces after 7 days of seeding. As shown in Figure 8a–e, a more intense blue color was noticed for cells cultured on CaP-coating surfaces compared to those on untreated Zn-0.5Li alloy. Maybe the low cell density on the untreated Zn-0.5Li alloy surface caused the low expression of ALP staining. ALP is an early marker for cell osteogenic differentiation. Quantitative testing of the ALP content of MC3T3-E1 cells cultured for 14 days is shown in Figure 8f. Compared to the cells cultured on TCP, the CaP coating

significantly enhanced the ALP activity of MC3T3-E1 cells. Although the MC3T3-E1 cells cultured on CaP-15 showed the highest average ALP activity, there was no significant difference among the CaP-coating groups (CaP-5, CaP-10, CaP-15, and CaP-60).

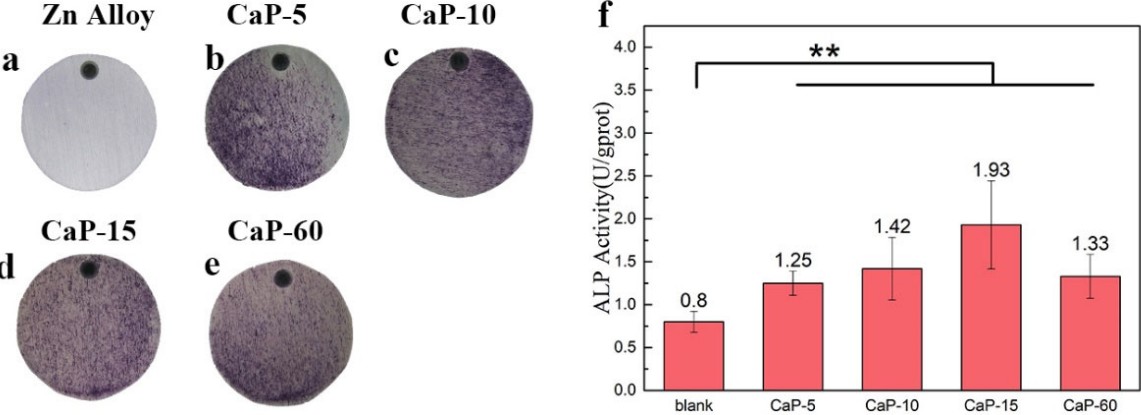

**Figure 8.** ALP staining results of MC3T3-E1 cells cultured on the Zn-0.5Li alloy surface with/without the CaP coating for 7 days: (**a**) untreated Zn-0.5Li alloy, (**b**) CaP-5, (**c**) CaP-10, (**d**) CaP-15, (**e**) CaP-60. (**f**) ALP activity of MC3T3-E1 cells cultured on different surfaces for 14 days. Statistical significance: ** $p < 0.01$.

## 4. Discussion

We investigated the calcium–phosphorus coating treatment on the surface of Zn-0.5Li alloy as a method to solve the problems of controlling the release of zinc alloy and increasing biological and osteogenic activity. Through XPS and XRD analyses, we determined that the coating consisted of $Zn_3(PO_4)_2$, $CaZn_2(PO_4)_2$, and $ZnLiPO_4$ compounds. SEM images demonstrated that the thickness of the coating increased with treatment time. When the coating thickness was below 10µm, a strong bond was observed between the coating and the substrate. However, when the coating exceeded 10µm, cracks appeared between the coating and the substrate as shown in the SEM images. The micron scratch test results also confirmed that the bond strength between the coating and the substrate decreased as the coating thickness increased. The phenomenon occurred because of the different crystal structures of the compounds in the CaP coating compared to the Zn alloys. The CaP coating exhibited a higher surface roughness than the Zn alloys. Among the samples, the CaP-10 sample with a coating thickness of less than 10µm showed the highest roughness of up to $0.575 \pm 0.074$µm. The rough surface provided adhesion points for cell growth, promoting cell adhesion and spreading on the coating surface, thereby enhancing the biocompatibility of the sample [23]. In the study of corrosion products and the efficiency of ion release from samples, we observed that zinc-containing corrosion products deposited the surfaces of the coated samples, and the coatings effectively prevented the occurrence of $Zn^{2+}$ burst release and pitting corrosion that happened during degradation of the bare metal. The degradation process of the coated samples proceeded at a stable rate, with the degradation rate decreasing as the coating thickness increased. In experiments evaluating cell activity, adhesion, and proliferation, we found that although cell density on CaP-5 and CaP-10 samples was lower compared to CaP-15 and CaP-60 samples, it was still higher and exhibited better adhesion than cells grown on zinc alloy surfaces. This indicated that the CaP coating can significantly improve the biocompatibility of zinc alloys. There was no significant difference in the ALP assay results among cells grown on coatings of different thicknesses, but all of them were higher than those grown on the zinc alloy surfaces, suggesting that CaP coatings can enhance the osteogenic activity of zinc alloys. Combining the results from our experiments, we can preliminarily conclude that CaP coatings effectively prevent initial ion burst and subsequent pitting corrosion, achieving controlled degradation of zinc alloys, and improve the bioactivity and osteogenic activity of

zinc alloys. Coatings with a thickness of less than 10 μm not only ensured a tight bond with the substrate but also possessed a rough surface. More importantly, the coatings ensured stable degradation of the substrate while avoiding pitting corrosion and burst release of $Zn^{2+}$ during the degradation process, improving the biocompatibility and osteogenic activity of zinc alloys.

As previously reported, the calcium–phosphorus coating can effectively control the degradation of a magnesium–calcium alloy during the preimplantation period, thereby extending the degradation time to solve the problem of excessive degradation rates of magnesium alloys [32]. The calcium–phosphorus coatings we obtained on zinc alloy surfaces were also able to achieve controlled degradation by avoiding initial burst release and later pitting corrosion. While Sachiko et al.'s [32] study used hydroxyapatite as the component of the calcium–phosphorus coating, our CaP coating consisted of $Zn_3(PO_4)_2$, $CaZn_2(PO_4)_2$, and $ZnLiPO_4$. Both studies demonstrated the beneficial effect of calcium–phosphorus coatings in achieving controlled release. However, Sachiko et al.'s hydroxyapatite coating did not effectively address the problem of pitting during the degradation of bare metal in body fluids or simulated body fluids. In contrast, the CaP coatings we obtained ensured stable substrate degradation while effectively preventing pitting corrosion [32]. Li et al.'s study concluded that coatings containing zinc phosphate and calcium phosphate components can enhance the biocompatibility of zinc alloys [20]. Our coatings also contained elements such as Ca and P, and the experimental results demonstrated improved biocompatibility of Zn-0.5Li alloys. This suggested that the presence of elements such as Ca and P in the coating promotes cell adhesion and growth. Our ALP assay for cells tentatively suggested that this CaP coating also had good osteogenic activity.

Related studies have shown that both the structure and chemical composition of CaP coatings played important roles in enhancing the bioactivity of zinc alloys [33,34]. CaP coatings had micro/nano-structured surfaces, and such surfaces enhanced cell adhesion, provided initial anchoring for cells, and modulated cell adhesion behavior. Additionally, the presence of calcium in the coating facilitated the adhesion of fibronectin and hyaluronan, crucial proteins that influence cell adhesion and spreading [20,35]. In terms of enhancing the osteogenic effect of zinc alloys, CaP coatings provided essential ions such as $Ca^{2+}$ and $PO_4^{3-}$ that played significant roles in cellular reactions. $Ca^{2+}$ not only promoted mineralization of the extracellular matrix but also activated various signaling pathways involved in osteogenesis, including the β-catenin pathway and the ERK pathway [36,37]. These pathways stimulated interactions between osteoblasts and osteoclasts, facilitating bone remodeling [37]. $PO_4^{3-}$, as an important signaling molecule, can influence a variety of biological processes related to osteogenesis, such as the expression of osteogenesis-related genes (e.g., osteoprotegerin) and the secretion of bone-related proteins (e.g., matrix Gla protein (MGP)) [38]. Through the combined effects of $Ca^{2+}$, $PO_4^{3-}$, and other ions, the CaP coating significantly improves the osteogenic activity of zinc alloys.

This is our preliminary study on coatings aimed at controlling the degradation of zinc alloys. The application of calcium–phosphorus coatings on the surface of Zn-0.5Li alloys effectively controlled the degradation rate of the zinc alloys and addressed issues of uncontrolled degradation such as burst release of $Zn^{2+}$ and pitting corrosion on the alloy surface. In in vitro experiments, the CaP coating showed good biocompatibility and osteogenic activity, but the in vivo situation such as the potential for delamination of the coating under dynamic physiological conditions that occurred in the loaded areas and the repair of bone need to be further investigated. While the coating improved the biocompatibility of the Zn-0.5Li alloy, it also reduced the degradation rate, which may not meet the degradation rate requirements of certain clinical applications. Ideally, a biocompatible coating would be able to intelligently promote or inhibit the degradation of the substrate based on specific degradation rate requirements. Our future research will focus on the development of controlled-release coatings that can better fulfill the needs of implantation.

## 5. Conclusions

In this work, different thicknesses of CaP coatings were prepared on the surface of Zn-0.5Li alloy using the chemical conversion method. The CaP coating was formed by irregular micro-particles of phosphate compounds, and Zn was one of the elements in addition to Ca and P. The thickness of the coating layer, coating morphology, and bonding strength of the coating and the substrate could be regulated by the treatment time during coating layer formation. The electrochemical experiments showed that CaP coating could improve corrosion resistance and inhibit the explosive release of zinc ions of the Zn-0.5Li alloy in SBF. With the CaP coating on the surface of the Zn-0.5Li alloy, MC3T3-E1 cells could directly attach onto the surface and showed spreading morphology. Moreover, the CaP coating could enhance cell viability and proliferation significantly compared to the untreated Zn-0.5Li alloy. The ALP activity of MC3T3-E1 cells cultured on the surface of CaP coatings was also significantly enhanced. Thus, the CaP coating on the Zn-0.5Li alloy could inhibit the explosive release of zinc ions, and enhance biocompatibility and osteogenic activity.

**Supplementary Materials:** The following supporting information can be downloaded at: https://www.mdpi.com/article/10.3390/coatings14030350/s1. Scheme S1. (a) Optical images of cast Zn-0.5Li alloys ($\times 500$), (b) optical images of Zn-0.5Li alloys in the rolled state ($\times 500$), (c) SEM image of the Zn-0.5Li alloy.

**Author Contributions:** Conceptualization, H.X. and Y.T.; Methodology, H.X. and Y.T.; Validation, H.X. and Y.T.; Formal analysis, H.X. and Y.T.; Data curation, H.X. and Y.T.; Writing—original draft, H.X., X.F. and H.Z.; Writing—review & editing, Z.S., S.Y. and L.W.; Supervision, S.Y. All authors have read and agreed to the published version of the manuscript.

**Funding:** This work was supported by the National Key Research and Development Program of China (Grants No.2023YFC2412300), Beijing Nova Program (2022 Beijing Nova Program Cross Cooperation Program), No. 20220484178 and the project was selected through the open competition mechanism of the Ministry of Industry and Information Technology of P.R. China (Biodegradable Zn alloys for interference screws in sports medicine, 2023.09–2026.09).

**Institutional Review Board Statement:** Not applicable.

**Informed Consent Statement:** Not applicable.

**Data Availability Statement:** Data are contained within the article and Supplementary Materials.

**Conflicts of Interest:** The authors declare no conflict of interest.

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
