# Peer review of "Enhanced Biocompatibility and Osteogenic Property of Biodegradable Zn-0.5Li Alloy through Calcium–Phosphorus Coating"

_coatings, doi:10.3390/coatings14030350_

Round 1

Reviewer 1 Report

Comments and Suggestions for Authors

The work is interesting and well organised. Physico-chemical investigations and biological tests have been carried out. The paper is nice, but additional improvements are needed.

Minor points.

Page 3 Line 109. Notification should be given to the alloy manufacturer.

Page 5. Line 209 The microparticle size of the deposited coating should be shown in addition to the coating thickness.

Page 6 Line 271. Considering the implant application, the thickness of CaP coating layer should be controlled less than 10 μm.  This statement should be discussed.

 Major point. 

The discussion section is missing in the paper. The comparison of the previous results of the chemical conversion method and the results obtained should be added. In addition, the different methods of CaP coating deposition could be considered and presented from the point of view of thickness, corrosion and biological properties. The scientific novelty should be highlighted and clearly presented in the paper.

Reviewer 2 Report

Comments and Suggestions for Authors

This manuscript describes a study focussed on improving the biocompatibility and osteogenic properties of zinc and its alloys for use in biological implants. However, the degradation of zinc in biological environments can lead to an undesirable increase in local ion concentration, and the bare surfaces of these metals may not be inherently compatible with biological tissues. To overcome these challenges, a calcium-phosphorus (CaP) coating is applied to a Zn-0.5Li alloy in the study. This type of coating is known for its biocompatibility and osteoconductivity and is therefore often used to improve the surface properties of metal implants.

The study uses a chemical conversion method to apply calcium-phosphorus (CaP) coatings to surfaces of Zn-0.5Li alloys. This approach is characterised by its potential to improve the biocompatibility and osteogenic activity of the alloy, making it a strong point in terms of innovation in materials science for biomedical applications.

The research thoroughly characterises the CaP coatings in terms of morphology, structure and bonding strength with the substrate using SEM, XRD and scratch tests, which provide a detailed understanding of the coatings. The electrochemical experiments show that CaP coatings significantly improve the corrosion resistance of the Zn-0.5Li alloy in simulated body fluid (SBF). The study emphasises the possibility of regulating the thickness, morphology, and adhesion strength of the coatings by adjusting the treatment time during the coating process. The in vitro studies with MC3T3-E1 cells show that the CaP coatings improve cell viability and proliferation and increase alkaline phosphatase (ALP) activity, suggesting increased osteogenic potential. The ability of the CaP coatings to inhibit the rapid release of zinc ions is an important finding that eliminates a common problem of biodegradable zinc alloys in terms of potential cytotoxicity due to high localised ion concentrations.

The paper has a solid experimental part, but the discussion and conclusion are very poor and need a thorough revision to be considered worthy of publication.

The document does not mention any long-term in vivo studies to evaluate the performance of the CaP-coated Zn-0.5Li alloy implants. Such studies are important to evaluate the long-term degradation behaviour, biocompatibility and osteointegration of the implants in a living organism, which may differ significantly from in vitro conditions.

The study could benefit from a comparative analysis with other biodegradable coatings or materials currently used in orthopaedic applications. This would provide a clearer picture of where the CaP-coated Zn-0.5Li alloy stands in comparison to existing technologies.

While the study addresses the adhesion strength of the coatings to the substrate, it does not explicitly discuss the potential for delamination of the coating under dynamic physiological conditions that occur in loaded areas. This aspect is crucial for orthopaedic implants that are exposed to mechanical stresses.

The paper suggests that CaP coating enhances osteogenic activity, but a detailed mechanism explaining how the coating affects cellular behaviour and osteogenesis is not provided.

Although the study notes the inhibition of the explosive release of zinc ions, a detailed analysis of the kinetics of ion release over time and its correlation with coating thickness and degradation rate would provide valuable insights into the controlled degradation of the implants.

Finally, the authors should explicitly address the limitations of the study, including any assumptions made during the research and the limitations of the methods. A discussion of the potential clinical translation of the results should also be included, considering regulatory pathways, scalability issues and any foreseeable challenges in clinical application.

Round 2

Reviewer 1 Report

Comments and Suggestions for Authors

The autors has responded my question.

Reviewer 2 Report

Comments and Suggestions for Authors

The authors have implemented the revisions suggested by the reviewers, improving the overall quality of the text, and addressing the problems identified. Recommendation for publication.